# Temporal Causal Discovery and Generative Prediction of Vehicular $CO_2$ emission

## Abstract

Global warming from greenhouse gas emissions is humanity's largest environmental hazard. Greenhouse gases, like $CO_2$ emissions from transportation, notably cars, contribute to the greenhouse effect. Effective $CO_2$ emission monitoring is needed to regulate vehicle emissions. Few studies have predicted automobile $CO_2$ emissions using OBD port data. For precise and effective prediction, the system must capture the underlying cause-effect structure between vehicular parameters that may contribute to the emission of $CO_2$ in the transportation sector. Thus, we present a causal RNN-based generative deep learning architecture that predicts vehicle $CO_2$ emissions using OBD-II data while keeping the underlying causal structure. Most widely used real-life datasets lack causal relationships between features or components, so we use our proposed architecture to discover and learn the underlying causal structure as an adjacency matrix during training and employ that during forecasting. Our framework learns a sparse adjacency matrix by imposing a sparsity-encouraging penalty on model weights and allowing some weights to be zero. This matrix is capable of capturing the causal relationships between all variable pairs. In this work, we first train the model with widely used synthetic datasets with known causal structure among variables, then we apply it to the state-of-the-art OBD-II dataset to find the internal causal structure among the vehicular parameters and perform causal inference to predict $CO_2$ emission. Experimental results reveal that our causal discovery and forecasting method surpasses state-of-the-art methods for the tasks of causal discovery in terms of AUROC, forecasting on multivariate causal time series data, and OBD-II dataset in terms of MMD, RMSE, and MAE. After successful completion, we will release the code (Code for review - https://anonymous.4open.science/r/causal-obd-co2-0A0C).

## 1 Introduction

In real-world scenarios, multivariate time series data used are generally ubiquitous in nature, for example, electroencephalogram signals (Isaksson et al., 1981), climate records (Runge et al., 2019), stellar light curves in astronomy (Huijse et al., 2012), stock prices from the stock market (Zhang et al., 2017) and so on. Common data-driven decision making tasks for time series data encompass discovering anomalies, forecasting, classification, and so on. One of the most prevalent methods in forecasting or predicting future values is time series forecasting, which utilizes observations from the past (Li et al., 2023). In the past few years, temporal forecasting models have been applied in different applications of vehicular technology in the transportation sector. One such application is the emission monitoring and prediction of carbon dioxide ($CO_2$) and other greenhouse gases such as nitrous oxide and methane (Melo et al., 2022; Bai & Sun, 2024). Carbon dioxide ($CO_2$) is the primary driver of the greenhouse effect, and it is the main emission produced by the electricity, transportation, and manufacturing industries. Among which, transportation has long been a significant factor in the generation of $CO_2$ emissions (Yoro & Daramola, 2020). An effective emission monitoring system is necessary to regulate and restrict emissions. Monitoring vehicular emissions is challenging due to the vast quantity and diverse range of vehicles. Furthermore, conducting additional individual testing on such a vast number of vehicles is impractical. Previous works (Zeng et al., 2016; Grote et al., 2018; Oduro et al., 2013; Zeng et al., 2015) have demonstrated that a

data-driven model is effective for estimating $CO_2$ levels. However, these models lack the ability to deduce hidden characteristics that are not directly seen from the available data.

An effective generative forecasting model for time series should be capable of modeling both the transition model $p(x_t|x_{1:t-1})$ and the joint distribution $p(x_{1:t})$ for each given $t$. While popular predictive frameworks like kernel adaptive filters (KAF) (Liu et al., 2008), deep state-space models (SSMs) (Rangapuram et al., 2018) and the fundamental autoregressive integrated moving average (ARIMA) offer various approaches to capture $p(x_t|x_{1:t-1})$ or $p(x_t|x_{t-\tau:t-1})$ within a window of length $\tau$, it is important to note that these models are deterministic mappers and not generative. Put simply, these models cannot produce new values for time series by randomly selecting from a manageable hidden distribution. In recent times, there has been a growing interest in finding causality from time series data. For instance, in the fMRI data, it is crucial to determine the causal dominations between activated areas of the brain (Deshpande et al., 2009). The causal graph presented in this study may potentially offer valuable insights into the diagnosis of psychological illnesses based on brain network analysis (Wang et al., 2020). Unlike purely statistical correlations, causality provides a deeper understanding of how variables influence one another over time. In the context of vehicular $CO_2$ emissions, understanding that "engine load" causes "$CO_2$ emissions" allows us to identify and prioritize variables for emission reduction interventions. This is particularly crucial in vehicular systems, where spurious correlations may lead to suboptimal decision-making.

Therefore, the need for an efficient data-driven $CO_2$ emission prediction system from the environmental perspective, as well as the need for a dependable generative forecasting model and the contemporary inclination towards causal inference, is inevitable, especially for temporal data. Understanding causal links between different vehicular parameters (e.g., engine load $\rightarrow$ $CO_2$ emissions) enables identification of actionable factors contributing to emissions, thereby aiding decision-making for emission control strategies. Through this work, we propose a novel temporal causal discovery and generative forecasting framework, which itself is responsible for finding the underlying causal structure among different pairs of variables, learning them in the form of an adjacency matrix, and employing the knowledge during inference. To the best of our knowledge, our proposed work is the first endeavor that uses a recurrent variational autoencoder (VAE), consisting of a single-head encoder and multihead decoder, which integrates temporal causal discovery and generative forecasting of vehicular $CO_2$ emissions, a unique combination not addressed in recent state-of-the-art works. Formally, given an instance from a M-dimensional time series $x_t = \{x_t^d \,|\, \forall d \in [1, M]\}$, our framework comprises a recurrent encoder and a multi-head recurrent decoder, where each head is dedicated for generating a certain dimension of $x_t$ (i.e., $x_t^d$, where $d \in [1, M]$). In the decoder, we apply a sparsity penalty to the input to the hidden state weight matrix. This penalty encourages the model to come up with a matrix $A \in \mathbb{R}^{M \times M}$ that represents the underlying Granger causal relations between the dimensions $x$ in a sparse manner. In addition, we present an error-compensation module that considers the immediate impact $\epsilon_t$ of one process without considering its history.

We conduct two-stage experiments. At first, we perform experiments with synthetic sequences. Regarding the forecasting of time series, we assess the similarity between the distribution of past observations and the distribution of predicted data using both qualitative and quantitative methods. Regarding causal discovery, we evaluate our identified causal network by comparing it to state-of-the-art methods that also seek to detect causality. In the next phase, we apply our model on the widely used open-source OBD-II dataset (Rettore et al., 2016; 2018) to discover the causal structure and enhance the prediction of $CO_2$ emissions from vehicles in real-time. As compared to other recent state-of-the-art techniques, our model has outperformed in all the tasks.

## 2 RELATED WORKS

The proposed research is situated at the convergence of various areas of study, integrating concepts from temporal autoregressive models, identifying Granger causal relationships, and variational inferencing based time series models in the field of automotive applications.

### 2.1 GENERATIVE MODELS ON TIME SERIES

For generating synthetic time series data, simple Generative Adversarial Network (GAN) framework(Goodfellow et al., 2014) has been proposed, which leverages the recurrent neural network

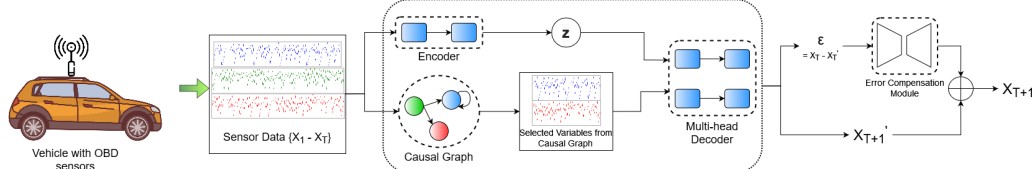

Figure 1: Overall working of our proposed model for vehicular $CO_2$ emission prediction. During inference, it uses the obtained causal graph during training to select the features which are affecting the $CO_2$ emission. The error compensation module attempts to estimate the errors made during forecasting and compensate that by adding with the output.

in modeling both the generator and discriminator (Mogren, 2016; Dimyati, 2021; Takahashi et al., 2019). The major drawback is that, these GAN-based techniques are only capable of modeling the joint distribution $p(x_{1:T})$, and they do not take into consideration the transaction dynamics $p(x_t|x_{1:t-1})$. This issue is addressed by TimeGAN (Yoon et al., 2019), which does so by estimating and integrating this conditional density knowledge inside an internal latent space. On the other hand, the variational autoencoder-based time series generation approach requires more exploration. Few of the notable works, such as one by (Fabius & Van Amersfoort, 2014) and Z-forcing(Goyal et al., 2017), have been proposed on this. However, the Z-forcing violates the fundamental Granger causality principle, which is cause precedes its consequence, by encoding the future information in the autoregressive structure. The recently introduced TimeVAE (Desai et al., 2021) makes use of convolutional neural networks in both the encoder and the decoder. Although it uses many concurrent blocks in the decoder, each taking into account a temporal aspect like trend and seasonality, it introduces additional hyperparameters that are challenging to measure in reality.

In the proposed framework, we have used a VAE-based generative forecasting framework; however, our approach differs in several factors from the baselines discussed above. It discovers the underlying Granger causal structure as a part of the training process, making it more transparent. It models the conditional density $p(x_t|x_{1:t-1})$ explicitly and guarantees the integration of learned Granger causal structure while generating the forecasted data from the encoded past observations.

## 2.2 CAUSAL DISCOVERY OF TIME SERIES

Various causal graphs may be explored for time series analysis (Assaad et al., 2022). In this context, we examine the process of recovering a Granger causal graph. This graph distinguishes between previous observations and current values of each variable and seeks to identify all potential causal relationships from the past to the present. Recent works have shown notable advancements in causal discovery and inference, following the principles of Granger causality (Granger, 1969).

The concept of 'causation' in time series was first introduced by Wiener (1956) in which, a time series or variable x is said to be the cause of another variable y if, from an analytical point of view, the value of y is enhanced by adding knowledge about x. However, Granger (1969) defined causality within the framework of linear multivariate auto-regression (MVAR) by contrasting the variances of the residual errors when x is included vs when it is excluded in the prediction of y. Later, the fundamental concept of Granger causality was introduced for non-linear scenarios by the use of the kernel technique (Marinazzo et al., 2008) and through the fitting of locally linear models in the reconstructed phase space (Chen et al., 2004). Recently, the first work leveraging deep neural networks to identify Granger causality has been carried out by (Tank et al., 2021), in which the causal graph is learned by the implementation of sparsity restrictions on the autoregressive network weights. The Temporal Causal Discovery Framework (Nauta et al., 2019) employs an attention mechanism that is embedded inside dilated depthwise convolutional networks in order to learn complicated non-linear causal links and, in certain instances, hidden confounders.

## 2.3 $CO_2$ EMISSION PREDICTION FROM OBD SENSOR DATA

The widespread implementation of cellular communication technology indicates that telematic applications can be easily accessed (Amarasinghe et al., 2015). This is the main factor driving the adoption of telematics-based prediction systems. Previous studies (Chen et al., 2015; Girma et al.,

Table 1: Comparison of tasks performed by the chosen baselines with our proposed approach.

| Model | Causal Discovery | Causal Inference | Data Generation |
|---|---|---|---|
| LPCMCI (Gerhardus & Runge, 2020) | ✔ | ✘ | ✘ |
| NGC (Tank et al., 2021) | ✔ | ✔ | ✘ |
| ACD (Löwe et al., 2022) | ✔ | ✘ | ✘ |
| PCMCI$_\Omega$ Gao et al. (2024) | ✔ | ✘ | ✘ |
| ScoreGrad (Yan et al., 2021) | ✘ | ✘ | ✔ |
| D$^3$VAE (Li et al., 2022) | ✘ | ✘ | ✔ |
| CSDI (Tashiro et al., 2021) | ✘ | ✘ | ✔ |
| Proposed | ✔ | ✔ | ✔ |

2019) have also shown the importance of telematics data, which can be employed in many industries and application areas. Moreover, with the progress in modern data-driven decision making technologies, telematics can make the systems highly significant and insightful. Earlier and recent works have employed artificial neural networks (ANN), support vector machines (SVM), VT-Micro approaches, and Bayesian neural networks to estimate $CO_2$ levels (Zeng et al., 2016; Nocera et al., 2018; Yavari et al., 2023). These works have used very limited number of features and have used straightforward non-temporal approaches for prediction, which have caused in poor estimation of $CO_2$ emission.

The work of Grote et al. (2018) includes Inductive Loop Detectors data and automotive sensor data including speed, acceleration, fuel flow, and mileage. The estimate is obtained from a hybrid model that uses the OBD model to set the parameters for the ILD model. Oduro et al. (2013) employed regression analysis to forecast $CO_2$ levels based on vehicle speed and acceleration, recommending emphasizing on OBD traits. Their investigation shows a direct relationship between $CO_2$ levels, velocity, and vehicle acceleration, with speed correlating more with emissions than acceleration. The study of Zeng et al. (2015) shows that factors other than vehicle speed and distance effect fuel use. The authors demonstrate a direct relationship between object velocity and fuel usage. Recently proposed work by (Singh & Dubey, 2021) uses long short term memory (LSTM) (Schmidhuber et al., 1997) based recurrent architecture for prediction of $CO_2$ emission from the OBD-II dataset. Their work uses statistical techniques such as correlation analysis and principal component analysis (PCA) to choose the features for training. This approach lacks in capturing the proper cause-effect relationships between the different OBD features.

The brief survey shows that $CO_2$ estimate and modeling approaches are very few in number and mostly use speed, congestion data, and specialized sensors. Although speed and acceleration are strongly linked to $CO_2$ emissions, they only provide limited insight into a vehicle's emission characteristics. Additionally, the survey reveals the utmost need of using a wide set of OBD features to accurately model $CO_2$ emission as the different factors are related to each other by some means. Therefore, there lies the possibility of strong spurious correlations among different OBD features, which may encode false or unwanted effects in forecasting (Calude & Longo, 2017; Yang et al., 2023). Thus, there is a strong need for integrating causal perspective and knowledge of cause-effect pairs of variables into the prediction model (Simon, 2017; Ye et al., 2024), which exactly we have done in this work. Causality provides a deeper understanding of the temporal influence of variables on each other as opposed to simple statistical correlations. Our proposed approach ensures to learn the effects of potential 'cause' variables on another variable in a pair-wise manner and apply that during inference to ensure that each variable in the generated instance should have a potential effect from its 'cause' variables only.

## 3 PROBLEM DEFINITION

The overarching goal is to acquire knowledge about a distribution $\hat{p}(x_{1:T})$ that is a close match to the actual joint distribution $p(x_{1:T})$. This is accomplished by sampling from a straightforward and controllable distribution $p(z)$ and then mapping to a more complex distribution $\hat{p}(x_{1:T})$. This is done from the standpoint of a generative model. In most cases, modeling $p(x_{1:T})$ is challenging due to the fact that its dimension M, length T, and perhaps non-stationary nature make it tough to work with. In order to derive the sequence in an iterative manner, we may utilize the autoregressive

decomposition, which is represented by the equation $p(x_{1:T}) = \Pi_{t=1}^{T} p(x_t|x_{1:t-1})$. Therefore, our first objective reduces to,

$$\min_{\hat{p}} \mathcal{D}(p(x_t|x_{1:t-1}) \parallel \hat{p}(x_t|x_{1:t-1})) \tag{1}$$

for any $t$, where $\mathcal{D}$ is the divergence between the two distributions. For the second objective, let us assume an instance from a M-dimensional time series be $x_t = \{x_t^1, x_t^2, \cdots, x_t^M\}$, which follows the Granger causal matrix given as $G = (V, E)$. An edge $(u, v)$ in the adjacency matrix $A$ of $G$ is given as,

$$A_{u,v} = \begin{cases} 1 & \text{if } x_{t-}^v \text{ causes } x_t^u \\ 0 & \text{otherwise} \end{cases} \tag{2}$$

where $x_{t-}$ denotes the previous observations of $x_t$. Let $PA(x^d)$ denotes the set of causes of $x^d$ in $G$. Integrating the additive noise model (Hoyer et al., 2008), $x^d$ can be represented as,

$$x_t^d = f_p(x_{t-}^d, PA(x^d)_{t-}) + \epsilon_t^d \tag{3}$$

where $PA(x^d)_{t-}$ represents the historical observations of the cause variables of $x_t^d$. Both $PA(x^d)_{t-}$ and $\epsilon_t^d$ are jointly independent with respect to $d$ and $t$, and for each $d$, they are independent and identically distributed in $t$. Consequently, our next objective is to ascertain $f_d(\cdot)$ for each $x^d$ and to determine the causal matrix $A$.

We then evaluate the approach on two synthetic causal sequences, Hénon and Lorenz-96, followed by applying the approach on OBD-II dataset. Experiments on synthetic causal sequences are important in order to show that our approach perfectly works for causal discovery as well as inferece, as OBD-II dataset does not have ground truths on causal relations.

## 4 PROPOSED METHODOLOGY

In this work, our main objective is to train a recurrent variational autoencoder which can approximate the actual joint distribution of the data and find the underlying Granger causal matrix. Furthermore, we apply the real world OBD-II dataset in order to learn its distribution, find the causal structure and forecast the emission of $CO_2$, following the discovered causal relations.

### 4.1 MODEL ARCHITECTURE

It consists of a RNN encoder and a multihead RNN decoder. The model with a past observation window size or lag $\tau$ can be mathematically denoted as,

$$\hat{x}_R = D_\Theta(x_R, E_\Phi(x_L)) + \epsilon_T \tag{4}$$

where $x_L = x_{T-2\tau-1:T-\tau-1}$, $x_R = x_{T-\tau:T-1}$, $\hat{x}_R = \hat{x}_{T-\tau+1:T}$, $D(\cdot)$ and $E(\cdot)$ represent the encoder and decoder, parameterized by $\Theta$ and $\Phi$ respectively. $\epsilon$ is the additive noise term which is assumed to follow no specific distribution. To simplify, given a time series $x = \{x_1, \cdots, x_\tau, \cdots, x_T\}$, the encoder takes the segment $x_{1:\tau-1}$ as input and passes it to the decoder. The decoder predicts the segment $x_{\tau+1:T}$, given the encoded latent vector $\mathbf{z} = E_\Phi(x_{1:\tau-1})$ and the segment $x_{\tau:T-1}$. In this manner, we adhere to the Granger causality principle by avoiding the encoding of future information prior to its decoding.

The model architecture block diagram has been given in Figure 2. Figure 2a shows the encoder block diagram of our model. Assuming $h$ be the hidden states of the encoder, we can mathematically represent it as,

$$\begin{aligned} h_t &= \tanh\left(W_{in}x_t + W_h h_{t-1} + b\right), \\ \mu &= W_\mu h_{T-\tau-1} + b_\mu, \\ \log(\sigma) &= W_\sigma h_{T-\tau-1} + b_\sigma \end{aligned} \tag{5}$$

where $\{W_{in}, W_h, W_\mu, W_\sigma\} \subseteq \Theta$. The weights associated with the input and hidden states are denoted with $W_{in}$ and $W_h$ respectively. $W_\mu$ and $W_\sigma$ are the weights used to calculate the mean and standard deviation of the derived Gaussian distribution, respectively. Bias is represented by $b$. Figure 2b displays the multihead decoder block diagram of our model. It illustrates the configuration of the first head utilizing a pentavariate slice of Hénon maps. The collection of all heads accurately represents the conditional probability distribution $p(x_t|x_{1:t-1})$. The decoder's starting state is randomly

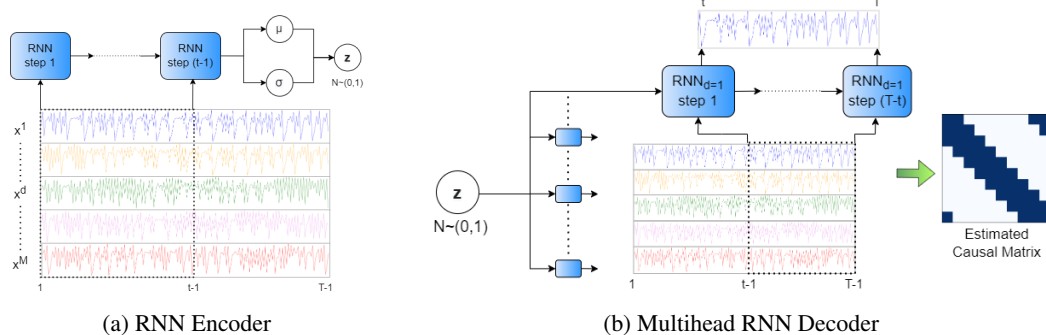

(a) RNN Encoder  (b) Multihead RNN Decoder

Figure 2: Block diagrams of Encoder and Decoder. We use $x_{0:t-1}$ segment as input to the encoder. The reparameterized latent vector $\mathbf{z}$ is passed to the decoder along with the segment $x_{t-1:T-1}$ in order to predict the segment $x_{t:T}$ sequentially. Each head of the decoder is responsible for predicting a specific dimension of $x$ i.e. $d = 1, \cdots, M$. During the training, the causal matrix is estimated in the form of adjacency matrix $\hat{A}$.

selected from a Gaussian distribution, which is defined by the parameters $\mu$ and $\sigma$. Considering $\mathbf{s}$ be the hidden state of the decoder, we can mathematically represent the decoder as,

$$\mathbf{s}_{T-\tau} = \tanh\left(U_{re}(\mu + \sigma\mathbf{z}) + b_{re}\right) \text{ where } \mathbf{z} \sim (0, 1),$$
$$\mathbf{s}_t^d = \tanh\left(U_{in}^d x_{t-1} + U_h^d \mathbf{s}_{t-1}^d + b^d\right),$$
$$\hat{A}_d = U_{in}^d,$$
$$\hat{x}_t^d = U_{out}^d \mathbf{s}_t^d + b_{out}^d$$

(6)

where $\{U_{in}^d, U_{re}, U_h^d, U_{out}^d\} \subseteq \Phi$. The weights associated with the input and hidden states of the $d^{th}$ decoder head are denoted as $U_{in}^d$ and $U_h^d$. Similarly, $U_{re}$ and $U_{out}^d$ denote the reparameterization and output layer weights respectively. $\hat{A}_d$ denotes the $d^{th}$ row of the estimated causal matrix $\hat{A}$ representing the Granger causal graph. The nonzero entries in $\hat{A}_d$ are the cause variables of $d^{th}$ variable.

Popular works on recurrent VAEs (Fabius & Van Amersfoort, 2014; Fraccaro et al., 2016; Goyal et al., 2017) employ the same time segment in both the encoder and the decoder, which allows them to encode future information in the recurrent structure. Thus, we have followed the idea proposed in T-forcing (Williams & Zipser, 1989), and other predictive autoregressive models (Litterman, 1986; Bengio et al., 2015), which make use of the actual data from the past in order to forecast the current value in the sequence. Another characteristic which differentiates our model from the above mentioned works is that our decoder possesses multiple heads, with the $p^{th}$ head being utilized to approximate $f_p(\cdot)$ in Equation (3). Subsequently, the complete vector array $x_t$ is constructed by aggregating the outputs of all M heads. A concise explanation of the term $D_\Theta(x_R, E_\Phi(x_L))$ is that it acquires the ability to estimate a collection of $\{f_d(\cdot)|p = 1, 2, \cdots, M\}$.

In order to further enhance the performance of sequence creation, an error compensation module is utilized to approximate a complementary noise term $\epsilon_t$ in Equation (4), with an assumption that its value cannot be inferred from its past observations. To implement, we have used another recurrent variational autoencoder, parameterized by $\{\xi, \zeta\}$, to estimate $\epsilon_{T-\tau:T}$. As it does not unravel the estimated causal network $\hat{A}$, we employ the same sequence for both encoder and decoder functions.

## 4.2 OBJECTIVE FUNCTION

To estimate the causal matrix $A$ during training, we have employed sparsification method exploited in many recent causal discovery works (Liu et al., 2020; Tank et al., 2021; Marcinkevičs & Vogt, 2021). It imposes a sparsity-inducing penalty on $\hat{A}$, assuming that the actual causal matrix $A$ is sparse. Thus, we train our model using stochastic gradient descent and proximal gradients in order to minimize the penalized objective function as below,

$$\mathcal{L}_m(\Theta, \Phi) = \sum_{d=1}^M [\mathbb{E}_{q_\Phi(\mathbf{z}|x_L)}[\log_{p_\Theta}(\hat{x}_R|x_R, \mathbf{z})]] - \mathcal{D}_{KL}(q_\Phi(\mathbf{z}|x_L)\|p(\mathbf{z})) + \lambda R(\hat{A}) \quad (7)$$

---

**Algorithm 1** Training pipeline of our proposed approach

---

**Require:** Time lag $\tau$ ; ISTA step size $\lambda$ and learning rate $\gamma$; initialize models $\mathcal{F}_{\Theta,\Phi}$ and $\mathcal{F}_{\xi,\zeta}$;
**Input:** The multivariate time series $\{x_t\}_{t=1}^T$ with $M$ dimensions;
**Output:** Estimated Granger causal matrix $\hat{A}$ and trained models $\mathcal{F}_{\Theta,\Phi}$ and $\mathcal{F}_{\xi,\zeta}$;
 1: **while** not converged or stopping criteria not met **do**
 2:     Extract $x_L = x_{T-2\tau-1:T-\tau-1}$ from $\{x_t\}_{t=1}^T$
 3:     Calculate the gradients of $\mathcal{L}_{m_{cvx}}$
 4:     Update $\mathcal{F}_{\Theta,\Phi}$ using SGD
 5:     Update $U_{in}$ using proximal operation (Equation 10)
 6: **end while**
 7: Stack $U_{in}$ to obtain estimated causal matrix $\hat{A}$
 8: Remove all zero edges in $U_{in}$ following $\hat{A}$
 9: **while** not converged or stopping criteria not met **do**
10:     Update $\mathcal{F}_{\xi,\zeta}$ by minimizing Equation 8 using Adam
11: **end while**
12: **return** $\hat{A}$ and trained models $\mathcal{F}_{\Theta,\Phi}$ and $\mathcal{F}_{\xi,\zeta}$

---

where $x_L$, $x_R$, $\hat{x}_R$ denote the same as in Equation (4) and $p(\mathbf{z})$ is a standard normal distribution. The first component of the loss function is the mean squared error (MSE) loss, which encourages the model to closely match the sample space. This is follwed by the KL divergence term, which ensures that the latent space follows a Gaussian distribution. Lastly, a regularization term $R(\cdot)$ on the estimated causal matrix $\hat{A}$, which promotes sparsity and is controlled by the hyper-parameter $\lambda$. Similarly, the error compensation network is trained by minimizing the following,

$$\mathcal{L}_\epsilon(\xi,\zeta) = \mathbb{E}_{q_\zeta(\mathbf{z}_\epsilon|\epsilon_{T-\tau:T})}(\log p_\xi(\epsilon_{T-\tau:T}|\mathbf{z}_\epsilon)) - \mathcal{D}_{KL}(q_\zeta(\mathbf{z}_\epsilon|\epsilon_{T-\tau:T})||p(\mathbf{z}_\epsilon)) \tag{8}$$

The updates made by Equation (8) does not affect the estimation of $\hat{A}$.

## 4.3 TRAINING PIPELINE AND OPTIMIZATION

The optimal selection for $R(\cdot)$ in Equation 7 is the $l_0$ norm, which quantifies the amount of non-zero components. However, the optimization of the $l_0$ norm in neural networks remains a difficult task. Therefore, we utilize the $l_1$ norm, which transforms Equation 7 into a standard lasso problem. The proximal gradient descent algorithm is often used for optimizing non-convex lasso objectives. Iterative shrinkage thresholding algorithms (ISTA) (Daubechies et al., 2004; Chambolle et al., 1998) with fixed step size are commonly used in practice. The thresholding feature in $U_{in}^d$ results in precise zero answers. More precisely, we begin the process of updating the weights $U_{in}^d$ iteratively, starting with the initial weights $U_{in}^d(0)$ by following Equation 9.

$$U_{in}^d(i+1) = \text{prox}_{\gamma,\lambda}(U_{in}^d(i) - \gamma\nabla\mathcal{L}_{m_{cvx}}(U_{in}^d(i))), \tag{9}$$

$$\text{prox}_{\gamma,\lambda}(U_{in}^d) = U_{in}^d(1 - \frac{\lambda\gamma}{||U_{in}^d||_F})_+ \tag{10}$$

Here $\text{prox}_{\gamma,\lambda}(\cdot)$ represents the proximal operator with a step size $\lambda$ and $\gamma$ is the learning rate. $\mathcal{L}_{m_{cvx}}$ refers to the convex component of the loss function, which corresponds to the first and second terms in Equation 7. The proximal step for applying the group lasso penalty on the input weights involves performing a group soft-thresholding operation on the input weights (Gong et al., 2013), which is given in Equation 10. Here, $||x||_F$ represents the Frobenius norm and $(x)_+ = \max(0, x)$. During the training process, three distinct optimization approaches are utilized: stochastic gradient descent (SGD) on all parameters of main model, proximal gradient on the weights of the decoder input layers ($U_{in}^d$) of the main model and Adam optimizer on the error compensation network. After training, we can acquire the estimated causal matrix by stacking $U_{in}^d$. During prediction, we encode the past observations in $\mathbf{z}$ and acquire the prediction error $\epsilon$, for each dimension, of the first predicted value from the last instance of past observation, and then input them into the decoders to build a time series of any window length in a step-by-step manner. The overall training pipeline has been given in Algorithm 1.

### 4.4 DATASETS

We methodically assess the efficacy of our proposed approach in both causal discovery and time series forecasting using two extensively utilized synthetic causal time series datasets. This is followed by applying our method to the real life OBD-II dataset. Later, we have shown that our approach surpasses in causal forecasting, as compared to the baselines, when applied to OBD-II dataset.

**Hénon maps:** We choose 10 interconnected Hénon chaotic maps (Kugiumtzis, 2013), where the actual causal relationship is $x^{i-1} \rightarrow x^i$. We create a total of 5,000 samples that comprise our training and evaluation data.

**Lorenz-96 model:** The Lorenz model (Lorenz, 1996) is a nonlinear model used to simulate climate dynamics. We simulate a 10 dimensional (p=10) model with the forcing constant set as 10. We generate 5,000 samples that comprise our training and evaluation data. The equations for data generation and other details regarding synthetic causal sequences have been given in Appendix A.

**OBD-II dataset:** The OBD-II dataset (Rettore et al., 2016; 2018) is a real life dataset, which offers real-time sensor data within the car, including Engine Load, Manifold Pressure, Fuel Trim, and Engine RPM, as well as diagnostic trouble codes for the vehicle. The emission of an internal combustion engine is closely correlated to its sensor data, such as RPM, load, throttle, and other factors. Hence, the OBD-II data provides information, either directly or indirectly, on the emission characteristics of a vehicle (Ortenzi & Costagliola, 2010). The details of the dataset and features selected for the experiment have been discussed in Appendix B. The implementation setup details have been given in Appendix E.

## 5 RESULTS AND EVALUATION

First, we compare and evaluate our proposed model with state-of-the-art causal discovery and temporal generative models. This is followed by applying the same baselines of both the categories on the OBD-II dataset.

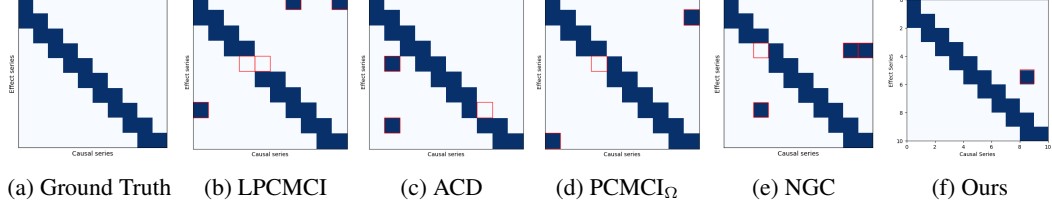

| (a) Ground Truth | (b) LPCMCI | (c) ACD | (d) PCMCI$_\Omega$ | (e) NGC | (f) Ours |
|---|---|---|---|---|---|

Figure 3: Estimated causal matrices from the chosen baselines for causal discovery and our proposed approach on Hénon maps. Wrongly identified causal relations have been highlighted using red rectangles.

### 5.1 EVALUATION ON CAUSAL DISCOVERY

As a baseline for Granger causal discovery, we have chosen four popular methods : LPCMCI (Gerhardus & Runge, 2020), a constraint-based causal discovery algorithm for autocorrelated time series with latent confounders that iteratively refines conditioning sets by including causal parents to improve the effect size of conditional independence tests and applies novel orientation rules for accurate identification of causal and ancestral relationships; NGC (Tank et al., 2021), the first neural network-based methods that actively and automatically extract causal relationships during the learning process; ACD (Löwe et al., 2022), a variational encoder-decoder framework that infers causal graphs from time-series data by leveraging shared dynamics across samples with different underlying causal graphs, enabling efficient and scalable causal discovery without refitting for each new sample; PCMCI$_\Omega$ (Gao et al., 2024), a non-parametric, constraint-based algorithm that extends the PCMCI framework to perform causal discovery in semi-stationary time series by detecting periodic changes in causal mechanisms through conditional independence tests. In addition to these, we have chosen three more causal discovery techniques closely relevant to our proposed approach, which have been discussed in Appendix D.1 due to space constraint.

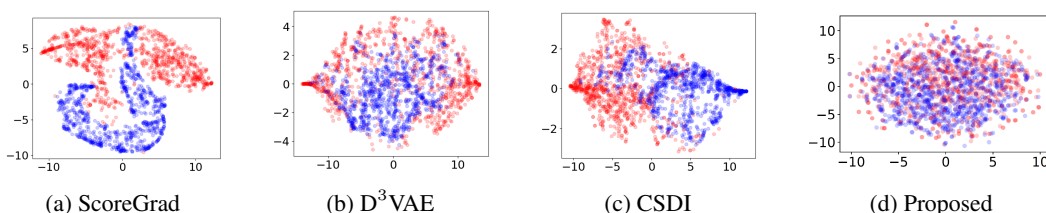

(a) ScoreGrad      (b) D³VAE      (c) CSDI      (d) Proposed

Figure 4: t-SNE visualizations on Hénon. Red samples represent actual time series $x_{t:T-1}$ and blue samples represent the forecasted time series $\hat{x}_{t+1:T}$.

In each technique, we assess the calculated causal adjacency matrices by comparing them to the ground truth. We use the AUROC score as a quantitative indicator. When using neural network-based methods, we choose the predicted causal matrices by searching for the minimum convex loss. The hyperparameters of all learnable models are kept as mentioned in the respective works.

First part of Table 2 presents a concise overview and comparison of the quantitative results for causal discovery. The performance of our model surpasses that of other baselines in all datasets and is comparable to that of NGC. Figure 3 represents a qualitative comparison on Hénon maps in terms of the estimated causal adjacency matrix. All the baselines perform well in discovering self-cause; however, as compared to our proposed approach, they discover additional false causal relations. Note that we have not reported any causal discovery results for the OBD-II dataset, as no ground truth is available for this dataset.

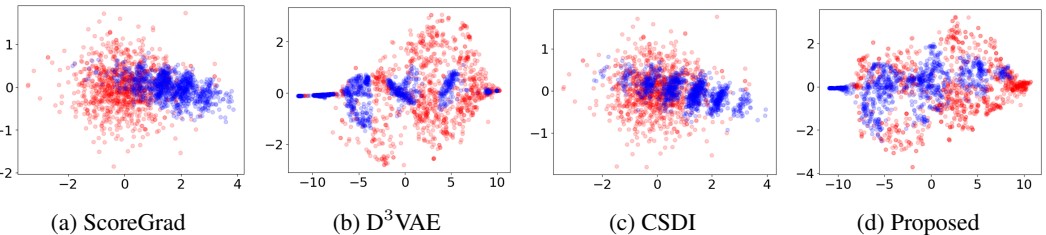

(a) ScoreGrad      (b) D³VAE      (c) CSDI      (d) Proposed

Figure 5: t-SNE visualizations on the OBD-II dataset. Red samples represent actual time series $x_{t:T-1}$ and blue samples represent the forecasted time series $\hat{x}_{t+1:T}$.

## 5.2 EVALUATION ON GENERATIVE TIME SERIES FORECASTING

As a baseline, we have chosen three baselines : ScoreGrad (Yan et al., 2021), a multivariate probabilistic time series forecasting framework that leverages continuous energy-based generative models, integrating a time series feature extraction module with a conditional stochastic differential equation-based score matching module to iteratively predict future states by solving reverse-time SDEs; D³VAE (Li et al., 2022), a generative forecasting model that leverages a coupled diffusion probabilistic process to augment time series data, multiscale denoising score matching to reduce noise and improve accuracy, and disentangled latent representations to enhance interpretability and robustness in forecasting; CSDI (Tashiro et al., 2021), which uses probabilistic forecasting, utilizing conditional denoising diffusion processes with self-supervised training and a two-dimensional attention mechanism to model future time steps based on historical data while capturing temporal and feature dependencies. We have modified their available implementations to generate, or rather forecast, the future predictions by taking the encoded latent variable generated from the past observations. In addition to these, we have chosen three more methods which are relevant to our proposed approach, which have been discussed in Appendix D.2 due to space constraint.

Initially, we assess the quality of the produced time series by subjectively evaluating both actual and forecasted ones using t-SNE (Van der Maaten & Hinton, 2008) by projecting them into a 2-dimensional space. An effective generative model is anticipated to promote the convergence of probability distributions for both actual and generated data. Figure 4 clearly shows that our proposed approach exhibits significantly higher overlap with the input data for Hénon maps as compared to ScoreGrad and CSDI, and it performs somewhat better than D³VAE. Figure 5 shows the

comparison when the methods are applied on OBD-II dataset. The predicted data can be seen creating clusters for all the approaches, however, in our proposed approach, the generated data clusters show better overlap with each other as well as the input data, as compared to the chosen baselines. Subsequently, we employ the maximum mean discrepancy (MMD) (Gretton et al., 2012) and root mean square error (RMSE) to quantitatively assess the effectiveness of various techniques. More precisely, MMD is employed to quantify the discrepancy between generated data and actual data. If a generative model accurately represents the underlying transition dynamics of a real time series (i.e., $p(x_t|x_{1:t-1})$), it is predicted to have minimal prediction error. As shown in Table 2, our proposed approach outperforms all the baselines in generative forecasting in terms of RMSE, while it is slightly outperformed by $D^3VAE$ in terms of MMD for Lorenz-96 data.

Table 2: Quantitative comparison for causal discovery and forecasting. Best results have been given in **bold**. The second best results have been underlined. In Simple RNN-based forecasting, LSTM refers to the work of Singh & Dubey (2021).

| Task | Metric | Model | Hénon | Lorenz-96 | OBD-II |
|---|---|---|---|---|---|
| Causal Discovery | AUROC | LPCMCI | 0.854 | 0.713 | NA |
| | | NGC | 0.961 | 0.961 | NA |
| | | ACD | 0.825 | 0.645 | NA |
| | | PCMCI$_\Omega$ | 0.907 | 0.827 | NA |
| | | Proposed | **0.980** | **0.975** | NA |
| Generative Forecasting | MMD | ScoreGrad | 0.395 | 0.109 | 3.225 |
| | | D$^3$VAE | 0.127 | **0.015** | 1.856 |
| | | CSDI | 0.195 | 0.076 | 1.976 |
| | | Proposed | **0.119** | 0.019 | **1.839** |
| | RMSE | ScoreGrad | 0.236 | 0.124 | 0.821 |
| | | D$^3$VAE | 0.139 | 0.119 | 0.529 |
| | | CSDI | 0.165 | 0.158 | 0.676 |
| | | Proposed | **0.120** | **0.109** | **0.492** |
| Simple RNN-based forecasting | RMSE | LSTM | 0.159 | 0.142 | 0.932 |
| | | Proposed | **0.120** | **0.109** | **0.492** |
| | MAE | LSTM | 0.136 | 0.129 | 0.816 |
| | | Proposed | **0.109** | **0.093** | **0.688** |

## 6 CONCLUSION

The study proposes a temporal causal recurrent framework that incorporates the principles of Granger causality from the data during training and incorporates the knowledge during forecasting. We have also shown how this model can be used for $CO_2$ emission prediction for vehicles using the open source real-life OBD-II dataset. We have performed experiments using two causal synthetic dynamic systems, namely Hénon and Lorenz-96, widely used for temporal causal discovery research. Our model has shown the lowest maximum mean discrepancy value for Hénon, the second lowest value for Lorenz-96 and outperformed all the chosen baselines in terms of prediction root mean square error. Furthermore, we trained all the baselines on OBD-II dataset and compared their forecasting performance with the proposed model. Again, we have achieved lowest value of root mean square error and mean average error.

Future temporal Granger causal discovery research should include unmeasured confounders affecting known variables. Traffic and road conditions affect OBD sensor readings and the model's $CO_2$ emission predictions, subject to vehicle environment changes. Thus, to better understand prediction algorithm outcomes from vehicular data, the recommended technique must be evaluated across different vehicle types and conditions.

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

# A SYNTHETIC DATA GENERATION AND SPLITTING

Following these equations, we have generated the Hénon chaotic maps,

$$x^1_{t+1} = 1.4 - (x^1_t)^2 + 0.3x^1_{t-1}, \tag{11}$$

$$x^d_{t+1} = 1.4 - (ex^{d-1}_t + (1-e)x^d_t)^2 + 0.3x^d_{t-1}, \tag{12}$$

where $d \in [2, M]$, $M$ is the dimensionality, $e = 0.3$ and $d = 10$. The true causal relations $x^d \to x^{d+1}$ and self causal relations $x^d \to x^d$ in the corresponding adjacency matrix should be 1. The lag length has been taken as 2.

Similarly, we have used the following equations for simulating the M-dimensional Lorenz-96 chaotic maps,

$$\frac{dx^p_t}{dt} = (x^{p+1}_t - x^{p-2}_t)x^{p-1}_t - x^p_t + F \tag{13}$$

where $x^{-1}_t = x^{M-1}_t, x^0_t = x^M_t, x^{M+1}_t = x^1_t$, $p$ is the index variable and $F$ is the forcing constant that dictates the degree of nonlinearity and chaos in the series.

We generated around 5000 samples for each dataset. For the Hénon and Lorenz-96 systems, we sample initial values from a typical Gaussian distribution and then infer the trajectories via transition functions.

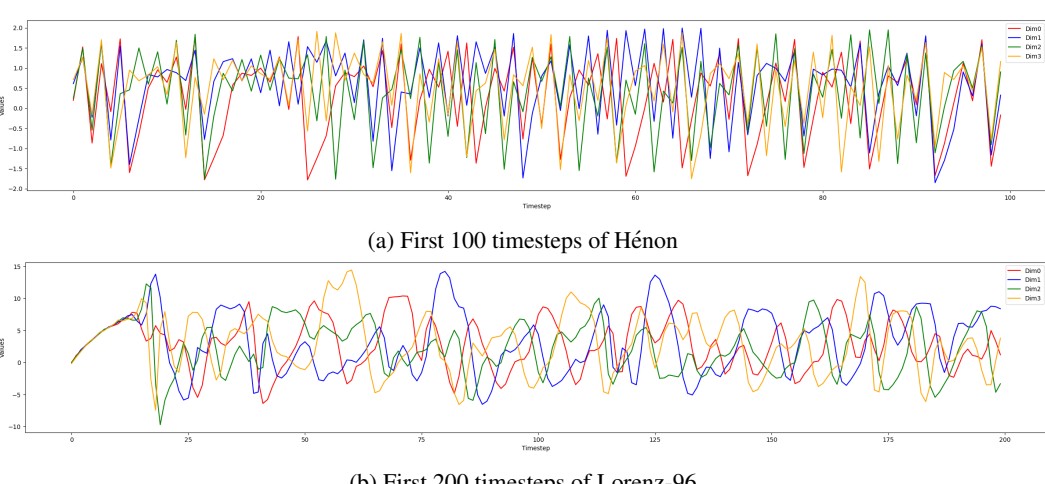

(a) First 100 timesteps of Hénon

(b) First 200 timesteps of Lorenz-96

Figure 6: Visualization of the first 4 dimensions of the generated Hénon and Lorenz-96 datasets

The generated data is then divided into several chunks of consecutive timesteps. Each of the chunks is considered as a single instance. We divide the set of instances into train and validation sets in a ratio of 80:20. For all the experiments, we have taken the chunk size $X_{1:T} = 30$. As shown in Figure 2, each chunk is then divided into two parts, $X_{1:\tau-1}$ and $X_{\tau:T-1}$ where $\tau = 15$.

# B DETAILS ON OBD-II DATASET

The OBD-II dataset was introduced in the work by Rettore et al. (2016). Mainly, the authors utilized an OBD Bluetooth adapter and a smartphone to collect data from two vehicles, then analyzed the correlation between RPM and speed data to ascertain whether it indicates the vehicle's present gear. The two cars belong to the same size category, although their manufacturers and engine power differ. The OBD-II interface, extensively utilized, was developed to standardize the physical connection, its pin configuration, signaling protocols, and message format. It is utilized for aftermarket maintenance, granting access to engine trouble codes and describing mechanics about breakdowns across the vehicle, therefore conserving significant time in diagnosing the source of issues.

The dataset contains around 43 features, including the car and driver identifiers, features directly obtained from the OBD scanner, calculated features, and variables obtained by smartphone sensors

such as GPS, Altitude and Air pressure. As our work focuses on vehicle sensor data, we have manually removed the features other than those directly obtained from the OBD sensors. We leave the investigation of causal effects of other variables in the dataset in $CO_2$ emission as a future work. The selected 15 OBD features, excluding the car identifier (Car ID), used in this work are summarised in Table 3.

Table 3: Summarization of the OBD features used in our study

| Feature name | Description |
| --- | --- |
| Intake Air Temperature | Temperature of the air utilized in the air-fuel combination. |
| Engine Temperature | Current temperature of the engine coolant fluid. |
| Engine load | Degree of stress or force exerted on an engine during operation. |
| Engine RPM | Engine's Revolutions per Minute. |
| Fuel flow | Instantaneous usage of fuel by the engine. |
| Fuel level | Current amount of fuel. |
| Instant Mileage | Immediate fuel use per kilometer. |
| Average Mileage | Average fuel use per kilometer for each log. |
| Speed | Speed reading from odometer. |
| Ambient Temperature | Temperature of air around the vehicle. |
| Air pedal | Percentage, indicating how fully the pedal is pressed. |
| Acceleration | Change of speed between two observations. |
| Air drag force | Measure of aerodynamic resistance that opposes a vehicle's motion. |
| Instant $CO_2$ emission | Instantaneous $CO_2$ emission reading. |
| Average $CO_2$ emission | Average $CO_2$ emission reading. |

The total number of instances in the dataset is 91,794 in which, 81,095 instances come from vehicle 1 (Car ID = 1) and the rest of the 6,699 instances come from vehicle 2 (Car ID = 2). In our experiment, we have used the instances of vehcile 1 exclusively as training set and that of vehicle 2 as validation set.

## C    THEORETICAL ASPECTS OF CAUSAL INFERENCE

In order to complement our experiments using real-life OBD-II dataset with no causal ground truths available, we delve deep into the theoretical aspects of the causal inference of our model. We have imposed sparsification on $U_{in}^d$, the weights associated with the input to hidden states of the $d^{th}$ decoder head. For a $D$ dimensional time series dataset, we obtain the estimated causal adjacency matrix $\hat{A}_{D \times D}$ by stacking all the $U_{in}^d$, where $d \in D$. This is followed by normalizing and thresholding the values of the matrix, where the negative values are zeroed and positive values are retained as 1's, converting it to a sparse square binary matrix.

In $\hat{A}$, the columns represent the 'cause' variables and the rows represent 'effect' variables. More specifically, the 1's in the $d^{th}$ column represent the variables, which are the 'cause' variables of the $d^{th}$ variable. Similarly, the 1's in the $d^{th}$ row represent the variables that are being affected by the $d^{th}$ variable. From the generative forecasting point of view, the former is the essential one as for generating each of the dimensions $d$ at time step $t$, the $d^{th}$ column of $\hat{A}$ filters out the variables that are the 'cause' of the $d^{th}$ variable and incorporates their past observations in the generating process of that specific dimension. Thus, our proposed methodology brings transparency during data generation process by maintaining the cause-effect relationships between pair-wise variables.

## D    ADDITIONAL BASELINE COMPARISONS

In addition to the comparison with the recent state-of-the-art methods (§5.1 and §5.2), we have chosen few other significant works closely related to our approach.

### D.1 CAUSAL DISCOVERY

For the task of causal discovery, we have chosen four popular methods : Kernel Granger Causality (KGC) (Marinazzo et al., 2008), which utilizes the kernel trick to extend Granger causality from linear to non-linear scenarios; Transfer Entropy (TE) (Schreiber, 2000), estimated using the matrix-based Rényi's $\alpha$-order entropy functional (Giraldo et al., 2014); Temporal Causal Discovery Framework (TCDF) (Nauta et al., 2019) incorporates an attention mechanism into a neural network. KGC and TE utilize information-theoretic measures, that is, focusing on independence or conditional independence, and employ postprocessing techniques such as hypothesis testing. On the other hand, TCDF and our proposed approach are neural network-based methods that actively and automatically extract causal relationships during the learning process.

The quantitative comparison has been done in terms of AUROC, which has been shown in the first part of Table 4. Figure 7 represents a qualitative comparison on Hénon maps in terms of the estimated causal adjacency matrix. It is to be noted that Neural network-based methods significantly outperform classical methods KGC and TE. This is because conventional methods lack the ability to identify self-cause.

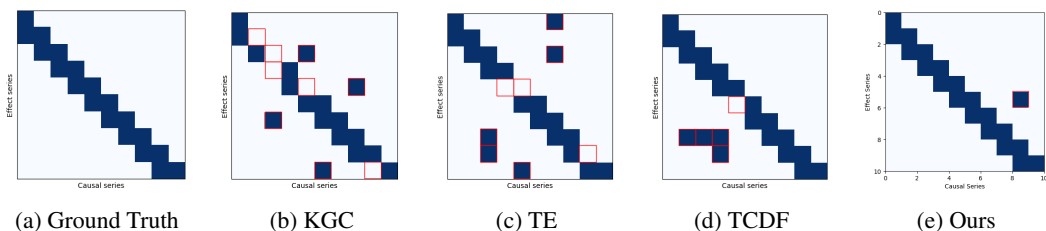

| (a) Ground Truth | (b) KGC | (c) TE | (d) TCDF | (e) Ours |

Figure 7: Estimated causal matrices from the additional baselines for causal discovery and our proposed approach on Hénon maps. Wrongly identified causal relations have been highlighted using red rectangles.

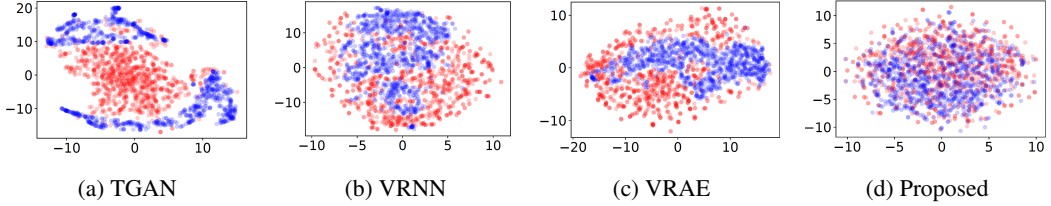

| (a) TGAN | (b) VRNN | (c) VRAE | (d) Proposed |

Figure 8: t-SNE visualizations on Hénon. Red samples represent actual time series $x_{t:T-1}$ and blue samples represent the forecasted time series $\hat{x}_{t+1:T}$.

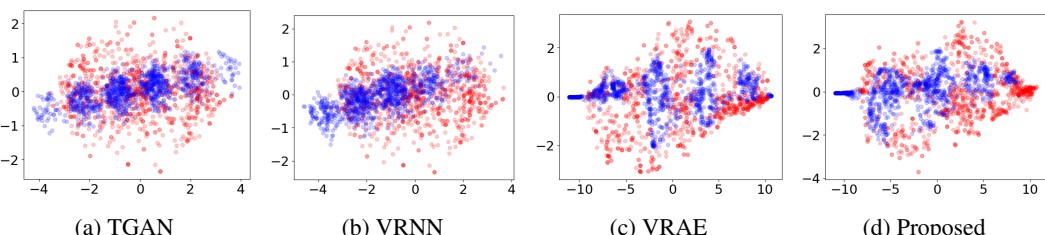

| (a) TGAN | (b) VRNN | (c) VRAE | (d) Proposed |

Figure 9: t-SNE visualizations on OBD-II dataset. Red samples represent actual time series $x_{t:T-1}$ and blue samples represent the forecasted time series $\hat{x}_{t+1:T}$.

Table 4: Quantitative comparison for causal discovery and forecasting. Best results have been given in **bold**. The second best results have been underlined.

| Task | Metric | Model | Hénon | Lorenz-96 | OBD-II |
|------|--------|-------|-------|-----------|--------|
| Causal Discovery | AUROC | KGC | 0.462 | 0.635 | NA |
| | | TE | 0.465 | 0.410 | NA |
| | | TCDF | 0.905 | 0.870 | NA |
| | | Proposed | **0.980** | **0.975** | NA |
| Generative Forecasting | MMD | TGAN | 0.475 | 0.039 | 2.109 |
| | | VRNN | 0.321 | 0.043 | 1.915 |
| | | VRAE | 0.125 | **0.011** | 1.877 |
| | | Proposed | **0.119** | 0.019 | **1.839** |
| | RMSE | TGAN | 0.291 | 0.124 | 0.901 |
| | | VRNN | 0.179 | 0.129 | 0.618 |
| | | VRAE | 0.125 | 0.123 | 0.756 |
| | | Proposed | **0.120** | **0.109** | **0.492** |

## D.2 GENERATIVE FORECASTING

For the generative forecasting task, we have chosen three baselines : Time-series generative adversarial network (TGAN) (Yoon et al., 2019) is a type of generative adversarial network that incorporates transition dynamics within the GAN framework; variational RNN (VRNN) (Chung et al., 2015) and variational recurrent autoencoder (VRAE) (Kingma, 2013). As these models are generative models, thus during inference, they generate data from a random noise sampled from a gaussian distribution $\mathcal{N} \sim (0, 1)$ (Brophy et al., 2023).

We assess the quality of the generated time series using t-SNE by projecting them into a 2-dimensional space. Figure 8 and Figure 9 show the qualitative comparisons with the above mentioned methods on Hénon maps and the OBD-II dataset respectively. The quantitative comparison has been done in terms of MMD and RMSE, which have been reported in the second part of Table 4.

## E IMPLEMENTATION SETUP

The whole system has been implemented in PyTorch v2.4.0 with support of CUDA v12.5 and trained on a single NVIDIA V100 GPU with 32GB VRAM. We have chosen single-layer LSTM (Schmidhuber et al., 1997) for implementing the main causal recurrent model and the error compensation model. We have set the value of $\lambda$ in PGD as 0.1 and kept the default values of $\beta_1$ and $\beta_2$ in Adam, which are 0.9 and 0.999, respectively. The maximum number of epochs is 2,000 and the learning rate $\gamma$ has been set at 0.03. Additionally, we have incorporated Early stopping regularization based on the validation loss, which stops the training if the validation loss does not decrease for 200 consecutive epochs.

