# Temporal Causal Discovery and Generative Prediction of Vehicular $CO_2$ emission

**Supplementary Material**

**Source code :**
The source code can be found here for review - https://anonymous.4open.science/r/causal-obd-co2-0A0C.

## 1 Design principles

There are a few unique designs that distinguish our proposed framework from the conventional recurrent variational autoencoder. These include the multi-head decoder, the unidirectional inputs, and the error compensation module.

### 1.1 Unidirectional inputs

The original VRAE, as well as its other notable variations (Goyal et al., 2017; Fabius & Van Amersfoort, 2014), employs the input of $x_{T-\tau:T-1}$ for both the encoder and the decoder. Before decoding, the information that pertains to the entire sequence is encoded in this manner. The estimation of $p(x_t|x_{1:t})$ is performed via these procedures, as opposed to $p(x_t|x_{1:t-1})$, which means that the future input values occurring at time $t$ cannot be utilized in the conditional variable. The concept that Massey et al. (1990) used to describe this phenomenon is known as causal conditioning. Due to the fact that it "peeps on the future," it is impossible for it to ever identify causality in the way that Granger (Granger, 1969) defines it. This is because it violates the fundamental rules of Granger causality within the context of causal discovery.

For the sake of providing evidence for our argument, we performed two separate experiments using a synthetic 10-dimensional linear autoregressive process with lag 3. Mathematically it can be written as,

$$x_t = \alpha_1 x_{t-1} + \alpha_2 x_{t-2} + \alpha_3 x_{t-3} + \epsilon_t \tag{1}$$

where $\epsilon_t \sim \mathcal{N}(0, 1)$. The true causal matrix for the synthetic process can be obtained by all non-zero elements of $\alpha_1 + \alpha_2 + \alpha_3$.

We choose to use the input of the encoder as $x_{T-\tau:T-1}$ rather than $x_{T-2\tau-1:T-\tau-1}$. As can be seen in Figure 1, our model is able to identify the majority of true causal relationships. On the other hand, the altered baseline, which has an encoder that looks at future values, is unable to identify the causal directions that exist between the majority of time series pairings.

### 1.2 Error Compensation Module

We subsequently validate the necessity of the error-compensation module. We analyze the time series generation outcomes of the original model and its inferior variant lacking error correction. We employ t-SNE (Van der Maaten & Hinton, 2008) to display the produced samples. An effective generative model should produce a distribution that closely resembles the real data distribution. Figure 2 illustrates that the error-compensation network results in a substantial increase in performance. Samples produced by the model without error correction rapidly converge to values approaching zero. We adjust the parameters $\{\alpha_1, \alpha_2, \alpha_3\}$ to prevent divergence, and the actual $x_t = \alpha_1 x_{t-1} + \alpha_2 x_{t-2} + \alpha_3 x_{t-3}$ did converge. The model without error compensation encapsulates the dynamics $p(x_t|x_{1:t-1})$, while disregarding $\epsilon_t$.

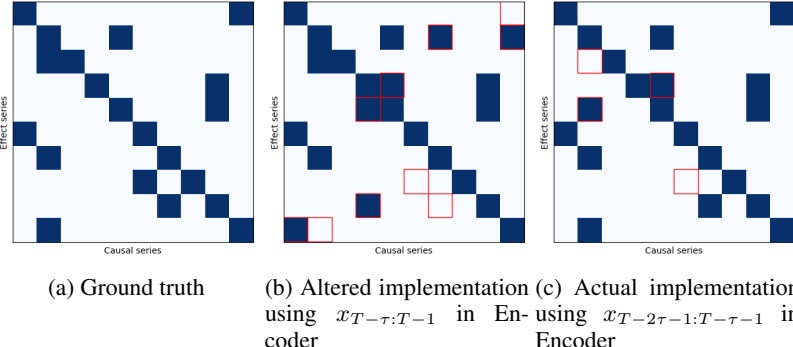

(a) Ground truth  (b) Altered implementation using $x_{T-\tau:T-1}$ in Encoder  (c) Actual implementation using $x_{T-2\tau-1:T-\tau-1}$ in Encoder

Figure 1: Visual comparison of causal graph adjacency matrices recovered by altered and actual implementation of our model with the ground truth causal matrix. Wrongly identified causal relations have been highlighted using red rectangles.

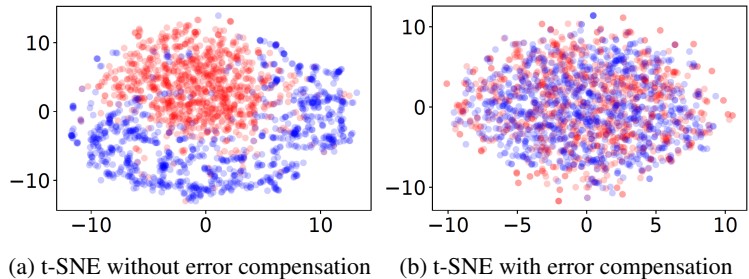

(a) t-SNE without error compensation  (b) t-SNE with error compensation

Figure 2: Visual comparison using t-SNE plots of the linear autoregressive process. Red samples represent original time series and blue samples represent the predicted time series.

The description and mathematical equations of multi-head decoder have been given in the main manuscript.

## 2 EVALUATION METRICS

Here we have provided a brief description and equations of the quantitative evaluation metrics we have used in our work.

**Area under Receiver Operating Characteristic curve (AUROC).** The Receiver Operating Characteristic (ROC) curve is a graphical representation that plots the True Positive Rate (TPR) against the False Positive Rate (FPR) at different threshold values. The AUROC represents the area under this curve, providing a single scalar value that summarizes the model's performance. The AUC ranges from 0 to 1, where, an AUC of 0.5 indicates no discriminative ability and an AUC of 1.0 indicates perfect discrimination between classes. It can be mathematically represented as shown in Equation 2.

$$\text{AUROC} = \sum_{i=1}^{N}(\text{FPR}_i - \text{FPR}_{i-1}).\text{TPR} \tag{2}$$

where TPR and FPR stand for True Positive Rate and False Positive Rate respectively and they can be given as,

$$\text{TPR} = \frac{\#\text{True Positives}}{\#\text{True Positives} + \#\text{False Negatives}}$$

$$\text{FPR} = \frac{\#\text{False Positives}}{\#\text{False Positives} + \#\text{True Negatives}}$$

We use the AUROC score as a quantitative indicator while assessing the calculated causal adjacency matrices by comparing them to the ground truth. The numerical results have been given in the main manuscript.

**Maximum Mean Discrepancy (MMD).** Maximum mean discrepancy (MMD) (Gretton et al., 2006) is extensively employed to quantify the divergence between two distributions. It assesses whether two sample sets—one synthetic and the other derived from real data—originate from the same distribution. We have used this method to quantitatively evaluate the chosen baselines and our model and compare the results in the task of generative forecasting. Formally it can be written as shown in Equation 3.

$$\text{MMD}(x, \hat{x}) = \frac{1}{n^2} \kappa(x_i, x_j) + \frac{1}{n^2} \kappa(\hat{x}_i, \hat{x}_j) - \frac{2}{n^2} \kappa(x_i, \hat{x}_j) \tag{3}$$

where $x$ and $\hat{x}$ are the actual data and the predicted data respectively, indexed by $i$ and $j$; the kernel $\kappa$ is designated as the Gaussian kernel. We compute the average results for a kernel size bandwidth of $[0.01; 0.1; 1; 10; 100]$, consistent with Goudet et al. (2018). The MMD provides greater informational value than both generator and discriminator loss in GAN-based models (Esteban et al., 2017), so we use early stopping to achieve the minimal MMD for generative models.

**Root Mean Squared Error (RMSE).** RMSE is one of the widely used metrics in forecasting models. It defined as the square root of the average of the squared differences between predicted and observed values. It essentially represents the standard deviation of the residuals, indicating how concentrated the data points are around the hyperplane of best fit. A lower RMSE value signifies a better fit, indicating that the model's predictions are closer to the actual outcomes. Equation 4 shows the mathetmaical representation.

$$\text{RMSE} = \sqrt{\frac{1}{N} \sum_{i=1}^{N} (y_i - \hat{y}_i)^2} \tag{4}$$

where $y_i$ is the ground truth, $\hat{y}_i$ is the predicted value and $N$ is the number of instances.

**Mean Average Error (MAE).** MAE is one of the widely used metrics in forecasting models. It is defined as the average of the absolute differences between predicted values and actual values. It quantifies how far off predictions are from the true values, providing a clear measure of prediction accuracy. The MAE is always non-negative, and a value of zero indicates perfect predictions. Lower MAE values suggest better model performance, while higher MAE values indicate larger discrepancies between predicted and actual values. MAE can be mathematically written as shown in Equation 5.

$$\text{MAE} = \frac{1}{N} \sum_{i=1}^{N} |y_i - \hat{y}_i| \tag{5}$$

where $y_i$ is the ground truth, $\hat{y}_i$ is the predicted value and $N$ is the number of instances.

## 3 RESULTS ON CAUSAL DISCOVERY

We have given the absolute adjacency matrices of Granger causal networks and match them with those calculated by our proposed approach and chosen baseline methodologies for causal discovery. In conventional Granger causality methods, we choose thresholds by optimizing AUROC and exclude any values below these thresholds. For neural network-based techniques, we may directly present the estimated matrices. As seen in Figures 3 and 4, our model surpasses the majority of baseline models and attains competitive outcomes among neural network-based approaches for both the Hénon and Lorenz-96 datasets. Red squares denote false positive or negative components.

## 4 RESULTS ON OBD-II DATASET

The obtained causal graph adjacency matrix from the OBD-II (Rettore et al., 2016) dataset has been shown in Figure 5. The rows and columns have been marked with their corresponding feature names. It can be clearly seen that our approach is able to retrieve all the self causal relations. Although we do not have ground truths available for the dataset, however, the cause variables for instantaneous $CO_2$ emission identified by the model are making sense. From the figure, the identified cause variables for $CO_2$ emission (apart from the instantaneous $CO_2$ emission variable itself) are - Engine load, Engine RPM, Instant mileage, Air pedal and Air drag force. All these variables somehow affect the vehicular $CO_2$ emission (Shiraki et al., 2020). As mentioned in the main manuscript, we have

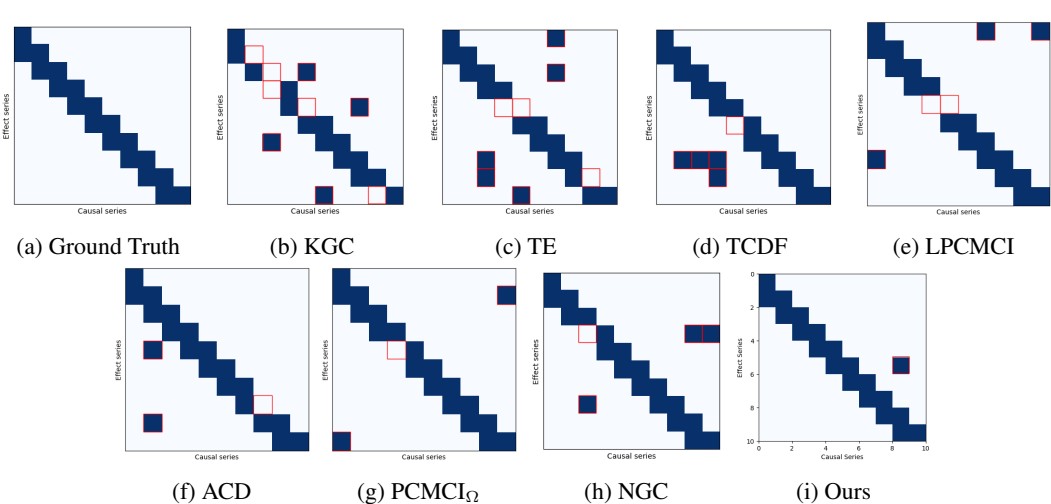

Figure 3: Estimated causal matrices from the chosen baselines for causal discovery and our proposed approach on Hénon maps. Wrongly identified causal relations have been highlighted using red rectangles.

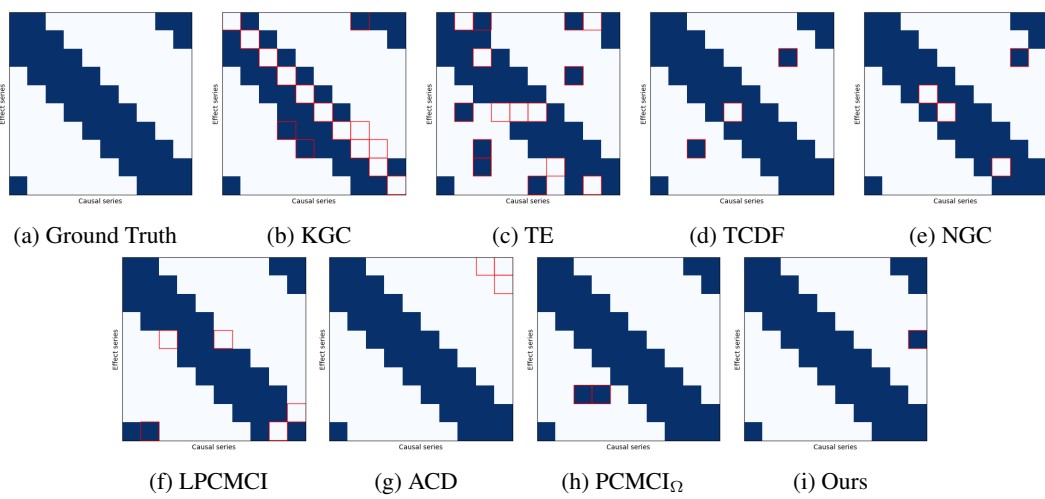

Figure 4: Estimated causal matrices from the chosen baselines for causal discovery and our proposed approach on Lorenz-96. Wrongly identified causal relations have been highlighted using red rectangles.

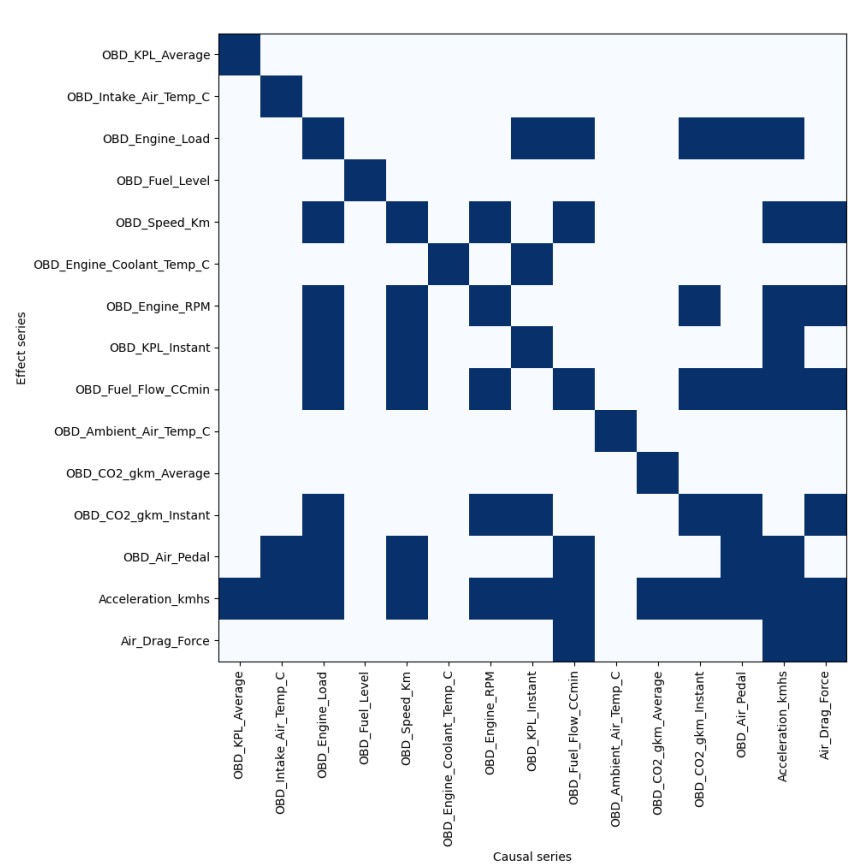

Figure 5: Retrieved Causal graph adjacency matrix of OBD-II dataset using our proposed approach

modified the existing implementations of the chosen generative model baselines to generate, or rather forecast the future predictions, by taking the encoded latent variable generated from the past observations. On the other hand, generative models generate data from a random noise sampled from a Gaussian distribution $\mathcal{N} \sim (0,1)$ during inference. We randomly sampled a chunk of 100 consecutive entries from the OBD-II dataset and applied these altered models as well as our proposed approach, followed by extracting the 'Instant $CO_2$ emission' feature values and plotting them, as shown in Figure 6.

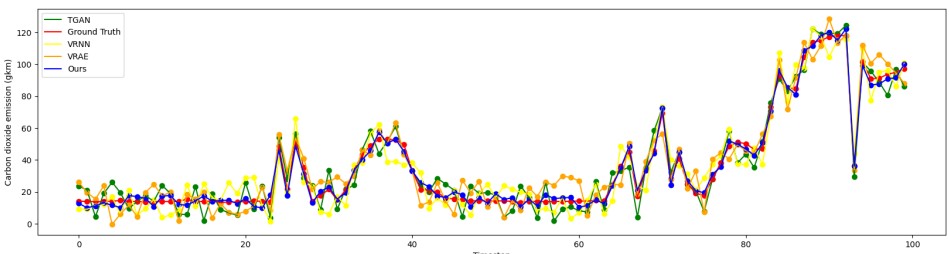

Figure 6: $CO_2$ emission prediction comparison with the chosen baselines and our model