# OpenReview forum: "Temporal Causal Discovery and Generative Prediction of Vehicular CO$_2$ emission"
_ICLR.cc/2025/Conference — Submitted to ICLR 2025_

### Official Review · Reviewer_D28V · 2024-10-29

**Soundness:** 2
**Presentation:** 2
**Contribution:** 2
**Rating:** 3
**Confidence:** 4

**Summary:**

This paper presents a new framework for predicting and discovering the casual structure of vehicular emission by learning a sparse adjacency matrix.

**Strengths:**

1.	The motivation of the paper is clearly presented, and the writing is easy to follow.
2.	The proposed method is tested on both real-world and synthetic datasets.

**Weaknesses:**

1.	Although the main focus of this paper is casual discovery, the casual discovery performance of the proposed method has not been tested on any real-world datasets.
2.	Some important implementation details are missing. For example, what are the exact form for the regularization term R in Eq. 7 and the forecasting lengths, receptively?
3.	The experimental results are weak. The proposed method should be compared to more state-of-the-art causal discovery methods in the field. Below are some suggestions:
a) Gao, Shanyun, Raghavendra Addanki, Tong Yu, Ryan Rossi, and Murat Kocaoglu. "Causal discovery in semi-stationary time series." Advances in Neural Information Processing Systems 36 (2023).
b) Löwe, Sindy, David Madras, Richard Zemel, and Max Welling. "Amortized causal discovery: Learning to infer causal graphs from time-series data." In Conference on Causal Learning and Reasoning, pp. 509-525. PMLR, 2022.
c) Gerhardus, Andreas, and Jakob Runge. "High-recall causal discovery for autocorrelated time series with latent confounders." Advances in Neural Information Processing Systems 33 (2020): 12615-12625.
4.	As a generative time series model, the proposed method also should be compared with state-of-the-art diffusion-based models, such as
a)	Tashiro, Yusuke, Jiaming Song, Yang Song, and Stefano Ermon. "Csdi: Conditional score-based diffusion models for probabilistic time series imputation." Advances in Neural Information Processing Systems 34 (2021): 24804-24816.

**Questions:**

1.	Why the results presented in Fig. 4 and 5 are used to indicate the effectiveness of the proposed method? The time steps of the data sample are different.

---

> ### Author Response · Authors · 2024-11-24
>
> Thank you for your valuable suggestions and specific questions, which open further detailed discussion.
>
> W1
>
> Our research corresponds with contemporary advanced methodologies that primarily assess causal discovery techniques using synthetic datasets with established ground facts. Real-world datasets, like the OBD-II dataset or any dataset pertaining to automobile emissions, lack definitive causal linkages among variables. Direct validation of causal discovery methods on large datasets is inherently problematic. Consequently, synthetic data, exemplified by the Henon maps and Lorenz-96 systems utilized in our research, are essential for verifying the precision of the causal discovery approach since these datasets offer a regulated environment with defined causal links. This guarantees that the causal discovery process undergoes thorough testing prior to its application in real-world contexts. As you proposed, we have been exploring and building our own dataset which includes causal ground truth pertinent to automotive applications, which we will utilize in our next projects.
>
> W2
>
> In Equation 7, the regularization term $\lambda R(\cdot)$ is the weighted ridge regularization term, which is applied to the $U_{in}^d$, where $d$ represents each dimension or variable of the dataset. Formally, it can be written as:
>
> $R (\cdot) = ||U_{in}||^2_F$
>
> where $\lambda$ is the coefficient term. This is applied to the input to hidden layer weights of each of the decoder heads.
>
> During training, we have created sliding windows of length $T$, each of which is divided into non-overlapping chunks, $x_L = x_{T-2\tau-1:T-\tau-1}$ and $x_R = x_{T-\tau:T-1}$ as the training chunks for encoder and decoder, respectively. The penalty is computed between $x_R = x_{T-\tau:T-1}$ and $\hat{x_R} = \hat{x}_{T-\tau+1:T} $.
>
> This prevents the model from encoding future information prior to decoding and violating Granger causality principles (Ref.: Section 4.1 Model Architecture). During inference, the forecasting length, i.e., the length of $x_R$, can be set by the user, as it does not overlap or depend on the length of $x_L$, the past observations.
>
> W3 & W4
>
> Thank you for your insightful input concerning the selection of baselines for the experimental evaluation of both causal discovery and time series generation. In accordance with your valuable ideas, we are presently bringing up the results and performing a comparative analysis with the recommended recent studies, which we will incorporate into the revised manuscript.
>
> Q1
>
> Thank you for raising this question. We appreciate the opportunity to clarify why the results in Fig. 4 and Fig. 5 are used to demonstrate the effectiveness of the proposed method. The t-SNE visualizations in Fig. 4 and Fig. 5 are included to qualitatively evaluate how well the generated time series samples align with the distribution of the corresponding input samples. We have treated each input chunk $x_L = x_{T-2\tau-1:T-\tau-1}$ of the time series data as an independent distribution and generated the corresponding samples $\hat{x_R} = \hat{x}_{T-\tau:T}$. Further, we have applied normalization to individual chunks and applied t-SNE to them. The resulting data was then plotted in a 2D space, with the input samples and generated samples represented using different colors (red for actual input samples and blue for generated samples). Moreover, we have generated the t-SNE plots during evaluation, and for every chunk, we have generated the plots by separate initializations. Thus, the axes in Fig. 4 and Fig. 5 do not represent time steps. This setup allows us to visually assess the overlap between the input and generated distributions.
>
> We hope to have addressed the reviewer's concerns. We welcome further suggestions to incorporate more justifications in the revised manuscript.

---

> > ### Comment · Reviewer_D28V · 2024-11-27
> > **Reply**
> >
> > Thank you for the feedback. If the ground truth for real-world data is not available, I would suggest that the author delve deeper into the theoretical aspects of causal inference. However, this is exactly the main weakness of the work: the lack of theoretical novelty in both causal discovery and forecasting. Therefore, my score will remain unchanged.

---

> > > ### Author Response · Authors · 2024-12-01
> > >
> > > Thank you for your reply. We would like to take another opportunity to elaborate about the theoretical aspect of how our proposed approach performs causal inference.
> > >
> > > In order to complement our experiments using real-life OBD-II dataset with no causal ground truths available, we delve deep into the theoretical aspects of the causal inference of our model. We have imposed sparsification on $U_{in}^d$, the weights associated with the input to hidden states of the $d^{th}$ decoder head. For a $D$ dimensional time series dataset, we obtain the estimated causal adjacency matrix $\hat{A_{D \times D}}$ by stacking all the $U^d_{in}$, where $d \in D$. This is followed by normalizing and thresholding the values of the matrix, where the negative values are zeroed and positive values are retained as 1's, converting it to a sparse square binary matrix.
> > >
> > > In $\hat{A}$, the columns represent the 'cause' variables and the rows represent 'effect' variables. More specifically, the 1's in the $d^{th}$ column represent the variables, which are the 'cause' variables of the $d^{th}$ variable. Similarly, the 1's in the $d^{th}$ row represent the variables that are being affected by the $d^{th}$ variable. From the generative forecasting point of view, the former is the essential one as for generating each of the dimensions $d$ at time step $t$, the $d^{th}$ column of $\hat{A}$ filters out the variables that are the 'cause' of the $d^{th}$ variable and incorporates their past observations in the generating process of that specific dimension. Thus, in our proposed methodology, it is distinctly visible which variable is affecting which variable during inference. This brings transparency during data generation process by providing the visibility as well as maintaining the cause-effect relationships between pair-wise variables.

---

### Official Review · Reviewer_21tT · 2024-11-04

**Soundness:** 3
**Presentation:** 2
**Contribution:** 4
**Rating:** 6
**Confidence:** 4

**Summary:**

The author proposes a recursive variational autoencoder (VAE) to approximate the joint distribution of time series data and improve the interpretability of time series forecasting by learning Granger causal structures. This framework also handles causal relationships within multivariate time series data and benefits future time step predictions through a generative model.

**Strengths:**

This paper offers an insightful approach by incorporating causal inference to enhance multivariate time series generation. For instance, this work could inspire future research in few-shot scenarios or cases where certain variables are sparse in multivariate time series generation. Additionally, it could delve deeper into identifying which variables contribute to long-term sequence modeling and which are more effective in capturing short-term temporal changes. I believe this paper has the potential to inspire a range of related future work.

**Weaknesses:**

1.Formatting Issue:
 Line 121 is missing a space.

2.Writing Issue:
First, the motivation for the work—improving multivariate time series forecasting by better learning causal relationships—has not been clearly communicated to the reader. It is important to show the potentially causal variables in the context of the CO2 forecasting problem. If text space is a concern, these details should be included in the appendix.
Additionally, if you can provide a brief explanation or demonstration of why understanding the intrinsic causal relationships in multivariate data leads to better generative time series forecasting, it would be a significant enhancement to the paper. Your current equations are close to achieving this, but the argument needs a bit more development.

3.Experimental Issue:
For the generative forecasting experiments, the authors compare TGAN (2019), VRNN (2015), and VRAE (2013). The choice of these baselines demonstrates the authors’ thoughtful consideration by directly comparing their work to the most relevant generative forecasting methods. However, in recent years, several emerging approaches for multivariate time series generative forecasting have been proposed, such as ScoreGrad (2021) and D3VAE (NeurIPS, 2022).
For simple RNN-based forecasting, I find that including only one RNN model as a baseline for non-generative multivariate time series modeling is insufficient. More recent and effective models should be considered in this context. Given the standards of ICLR, the experimental setup presents notable shortcomings.

**Questions:**

1. A statement is not strict enough：
The claim that the work you're referring to is the first to use a recurrent variational autoencoder (VAE) for time series forecasting is not entirely accurate. Previous research has already explored using Variational Recurrent Autoencoders (VRAE) for time series data. One notable example is the VRAE model (2014), VRNN (2015). The novelty of your mentioned work could lie in the specific architecture or methodology used, such as the incorporation of causal discovery through Granger causality and the multi-head decoder design, which may not have been part of earlier models. Could you further improve your claim about this?
2. Could you attempt to briefly demonstrate or explain why a better understanding of the intrinsic causal relationships within multivariate data can lead to improved multivariate time series generation?
Since the experimental results are not particularly strong, clarifying the rationale and theory behind your approach would help improve the overall quality and coherence of the paper.
3. As for supplementing the experiments, I recommend following the suggestions and opinions of other reviewers, especially about the multivariate time series model forecasting besides RNN.  I may not increase or decrease the score based on whether additional experiments are included in rebuttal phase.  The second item is more important for me.

---

> ### Author Response · Authors · 2024-11-24
>
> Thank you for your detailed and insightful comments.
>
> W1
>
> Thank you for pointing out this formatting issue. We have carefully reviewed the manuscript and corrected the missing space at Line 121. Additionally, we will conduct a thorough proofreading of the entire document to ensure that no similar formatting or typographical errors remain.
>
> W2
>
> 1. Thank you for highlighting this important aspect. Causality offers a more profound comprehension of the temporal effect of variables on one another, in contrast to mere statistical correlations. Recognizing that "engine load" contributes to "CO$_2$ emissions" enables us to discern and prioritize factors for interventions aimed at emission reduction. This is especially vital in vehicle systems because spurious correlations might result in inefficient decision-making. Our work explains causal relationships in OBD-II sensor data (e.g., engine load → CO$_2$ emission), facilitating the identification of actionable aspects that contribute to emissions and enhancing decision-making for emission control measures.
>
> 2. We acknowledge that demonstrating probable causative factors is essential in the context of vehicular CO$_2$ forecasting. However, with reference to this, we have already included causal discovery findings from the OBD-II dataset in the supplementary material. This study demonstrates the causative linkages between vehicle characteristics and their effect on CO$_2$ emissions, highlighting the method's significance and applicability in real-world contexts.
>
> 3. During training, we have used sparsity-inducing penalties on the input-to-hidden layer weights of each decoder head so that it can learn the underlying Granger causal relations among pairwise variables.
>
> W3
>
> 1. Thank you for your constructive feedback regarding the selection of baselines for the experimental evaluation. We appreciate your acknowledgment of our thoughtful inclusion of relevant generative forecasting methods. However, as per your insightful suggestions, we are currently in the process of reproducing the results and performing the comparative analysis with ScoreGrad and D3VAE, which we will include in the updated manuscript.
>
> 2. We understand your concern about including only one RNN-based model as a baseline for non-generative multivariate time series modeling. However, we note that simple RNN-based forecasting works for vehicular CO$_2$ emissions are very limited. The work by Singh et al. (2021) [1] is the most recent and relevant study directly aligned with our objective of vehicular CO$_2$ emission prediction. This work uses LSTM-based RNNs in conjunction with statistical feature selection techniques (e.g., correlation analysis and PCA).
>
> Q1
>
> Thank you for pointing this out. We acknowledge that previous works, such as VRAE (2014) and VRNN (2015), have explored the use of recurrent variational autoencoders for time series data. In contrast, we have developed a variational autoencoder (VAE), consisting of a single-head encoder and multihead decoder, which integrates temporal causal discovery and generative forecasting of vehicular CO$_2$ emission, a unique combination not addressed in previous works. Each head of the decoder is responsible for generating the data for each dimension of the forecasted time series. We have applied the sparsification trick to the input-to-hidden layer weights of the decoder heads, which helps the model learn the underlying Granger causal relations from the data during training. As per your valuable suggestion, we have revised our claim in the main manuscript in order to reflect the novelty of our work and differentiate it from earlier models more accurately.
>
> Q2
>
> As mentioned earlier, causality provides a deeper understanding of the temporal influence of variables on each other as opposed to simple statistical correlations. In our model, we have one decoder head dedicated to each of the variables/dimensions of the data. We have applied a sparsity-inducing penalty on the input-to-hidden layer weights of each decoder head, which finally makes some weights 0, mitigating the effect of the predictor variables that are not causally related to the response variable. This ensures that during inference in both forecasting and/or generating tasks, the trained sparse decoder head for the $i^{th}$ variable (dedicated to dimension $d_i$) will take into account the effects of potential cause variables on $d_i$. This approach mitigates the effect of "all" variables, therefore removing the effects of spurious correlations in the data-generating process and making the process transparent.
>
> We hope to have addressed the reviewer’s queries. We welcome further suggestions to incorporate more justifications in the revised manuscript.
>
> References :
>
> [1] M. Singh and R. K. Dubey, "Deep Learning Model Based CO2 Emissions Prediction Using Vehicle Telematics Sensors Data," in IEEE Transactions on Intelligent Vehicles, vol. 8, no. 1, pp. 768-777, Jan. 2023, doi: 10.1109/TIV.2021.3102400.

---

### Official Review · Reviewer_6wAX · 2024-11-04

**Soundness:** 3
**Presentation:** 3
**Contribution:** 3
**Rating:** 5
**Confidence:** 3

**Summary:**

The paper proposes a causal RNN-based generative deep learning architecture that predicts vehicle CO2 emissions using OBD-II data while maintaining the underlying causal structure. Unlike traditional models that miss hidden cause-effect relationships, this framework discovers and learns these relationships as an adjacency matrix during training by applying a sparsity-encouraging penalty. It captures causal links between variables and employs this knowledge for forecasting.

**Strengths:**

1. The background of this work is clearly presented, emphasizing that controlling CO2 emissions is a critical issue that needs to be addressed.
2. This paper designs and conducts two distinct experimental setups for causal discovery and emission forecasting, along with comprehensive result analysis.
3. The paper provides open-source code to enable the replication of its results.

**Weaknesses:**

1. The paper is not well written. The introduction does not summarize the contributions of this work. The importance of causal discovery for emission forecasting needs further clarification, including why causality between time series is significant for emission forecasting. For example, the causal analysis for traffic forecasting can help traffic managers know the reason for the jam happening.

2. The proposed method lacks novelty. The approach does not include designs specific to CO2 characteristics and relies solely on existing RNN-based architectures. Why do current approaches for time series forecasting struggle with accurately predicting CO2 emissions?

3. The selection of baselines for the forecasting task is insufficient, as it lacks recent state-of-the-art methods from the past three years.

4. While causal discovery appears effective, it lacks validation with real-world data. Case studies are needed to demonstrate its practical application. You can do case studies on the real-world dataset you provide, and show the discovery results.

**Questions:**

Same as the weakness.

---

> ### Author Response · Authors · 2024-11-24
>
> Thank you for your comments and highlighting some crucial points to be discussed in detail.
>
> W1
>
> 1. Our work proposes a variational autoencoder (VAE), consisting of a single-head encoder and multihead decoder, which integrates temporal causal discovery and generative forecasting of vehicular CO$_2$ emission, a unique combination not addressed in previous works.
> 2. Unlike purely statistical correlations, causality provides a deeper understanding of how variables influence one another over time. For example, understanding that "engine load" causes "CO$_2$ emissions" allows us to identify and prioritize variables for emission reduction interventions. This is particularly crucial in vehicular systems, where spurious correlations may lead to suboptimal decision-making. In our study, understanding causal links in OBD-II sensor data (e.g., engine load $\rightarrow$ CO$_2$ emission) enables identification of actionable factors contributing to emissions, thereby aiding decision-making for emission control strategies.
>
> Moreover, insights derived from causal discovery can inform regulatory policies or design interventions, such as optimizing vehicle maintenance schedules or designing emission control systems, which directly target causally significant parameters. We have revised the introduction, which explicitly mentions all these important points and specifically how these causal insights enhance forecasting and practical emission control.
>
> W2
>
> 1. We respectfully argue that while our framework incorporates existing RNN-based architectures, its novelty lies in its integration of causal discovery principles into the generative forecasting process.
>
> - Our work goes beyond conventional RNN-based forecasting by discovering Granger causal relationships between variables during training. This causal perspective ensures the model focuses on causally relevant features, which are not addressed by existing forecasting models.
>
> - We design and incorporate a sparsity-promoting penalty to encourage the discovery of sparse causal relationships, which is both computationally efficient and interpretable. This design specifically targets multivariate time series with interdependent variables like those in CO$_2$ emission data.
>
> - To address errors inherent in time series forecasting, we propose an additional module that estimates and compensates for residual errors, further improving prediction accuracy. This combination of causal discovery and error compensation is unique and tailored for complex multivariate systems.
>
> 2. We agree that it is crucial to understand the limitations of current approaches to emphasize the necessity of our work.
>
> - Traditional forecasting methods (e.g., ARIMA, LSTMs) rely on correlations between variables without uncovering causal relationships. In CO$_2$ emission prediction, where multiple factors (e.g., engine RPM, load, fuel flow) interact in non-linear and interdependent ways, ignoring causality can lead to spurious correlations and unreliable forecasts.
>
> - Real-life vehicular datasets like the OBD-II dataset used in existing works as well as in our work contain a large number of interdependent variables. Without causal guidance, models risk overfitting to noise or spurious patterns, especially in high-dimensional settings, which degrades their ability to generalize to unseen conditions. RNN-based existing approaches fail to capture these intricate dependencies effectively, leading to suboptimal predictions. Moreover, these works do not incorporate domain-specific knowledge, such as the known physical relationships between vehicular parameters (e.g., higher engine load leading to higher emissions). This results in limited interpretability and practical utility for emission management.
>
> In the revised introduction, we have addressed these gaps in existing methods in more detail, clarifying how our proposed framework overcomes them through its causal discovery and forecasting capabilities.
>
> (continued in the next comment)

---

> > ### Author Response · Authors · 2024-11-24
> >
> > (continuation)
> >
> > W3
> >
> > Thank you for highlighting the need to include more recent state-of-the-art methods as baselines in our forecasting task. We acknowledge the importance of benchmarking our proposed approach against the latest advancements to provide a comprehensive evaluation. We have incorporated the mentioned recent methods in our comparative analysis: Yan et al. [1], Li et al. [2].
> >
> > W4
> >
> > We agree that demonstrating the practical applicability of causal discovery through real-world case studies would enhance the impact of our work. However, as noted in the supplementary material, we have already included causal discovery results on the OBD-II dataset. In these results, we show the underlying causal relationships between vehicular parameters and their impact on CO$_2$ emissions, which highlights the method’s relevance and utility for real-world scenarios.
> > Our work aligns with recent state-of-the-art approaches, which predominantly evaluate causal discovery methods on synthetic datasets with known ground truths. Real-world datasets, such as the OBD-II dataset or any other dataset on vehicular emission application, do not come with ground truth causal relationships among variables. This makes direct validation of causal discovery methods on such datasets inherently challenging. Thus, synthetic data, such as the Henon maps and Lorenz-96 systems used in our work, are crucial for validating the accuracy of the causal discovery method because these datasets provide a controlled environment with established causal relationships. This ensures that the causal discovery mechanism is rigorously tested before applying it to real-world scenarios. However, as suggested by you, we are in the process of searching as well as creating our own dataset with causal ground truth related to vehicular applications, which we will use in our future works.
> > We hope to have addressed the reviewer's concerns. We welcome further suggestions to incorporate more justifications in the revised manuscript.
> >
> > References :
> >
> > [1] Yan, T., Zhang, H., Zhou, T., Zhan, Y., and Xia, Y., 2021. Scoregrad: Multivariate probabilistic time series forecasting with continuous energy-based generative models. arXiv preprint arXiv:2106.10121.
> >
> > [2] Li, Y., Lu, X., Wang, Y., and Dou, D., 2022. Generative time series forecasting with diffusion, denoise, and disentanglement. Advances in Neural Information Processing Systems, 35, pp. 23009-23022.

---

### Author Response · Authors · 2024-12-01
**General comments to all Reviewers**

Thanks to all the reviewers for their valuable comments and insightful suggestions. We have incorporated the suggestions and have submitted the revised manuscript. Here are the changes we have made:

1. Inclusion of baselines:

We have incorporated the results from the suggested baselines on causal discovery ([1], [2], [3]) as well as data generation ([4], [5], [6]) and compared the results with the ones of our proposed approach (Ref. - Section 5 'Results and Evaluation', Page 8-10). To maintain uniformity, we have applied the suggested baselines on the chosen datasets (H$\acute{\text{e}}$non, Lorenz-96, and OBD-II [7]) and evaluated in terms of the chosen metrics (AUROC for causal discovery; MMD and RMSE for generative forecasting).

2. Importance of Causality:

Unlike purely statistical correlations, causality provides a deeper understanding of how variables influence one another over time. In the context of vehicular CO$_2$ emissions, understanding causal links between different vehicular parameters (e.g., engine load $\rightarrow$ CO$_2$ emissions) enables identification of actionable factors contributing to emissions, thereby aiding decision-making for emission control strategies. This is particularly crucial in this field, as spurious correlations may lead to suboptimal decision-making (Ref. Section 1 'Introduction', Page 2).

3. Theoretical aspects of Causal inference:

We have made an attempt to complement our experiments using real-life OBD-II dataset with no causal ground truths available through explaining the theoretical aspects of causal inference performed by our model (Appendix C, 'Theoretical Aspects of Causal Inference', Page 16).

4. Additional baselines:

To further strengthen our paper, we have kept the existing baseline comparisons in the appendix instead of the main manuscript (Appendix D, 'Additional Baseline Comparisons'). Section 5 of the main manuscript contains qualitative and quantitative comparisons with the suggested baselines by reviewers.

References :

[1] Andreas Gerhardus and Jakob Runge. High-recall causal discovery for autocorrelated time series with latent confounders. Advances in Neural Information Processing Systems, 33:12615–12625, 2020.

[2] Sindy Löwe, David Madras, Richard Zemel, and Max Welling. Amortized causal discovery: Learning to infer causal graphs from time-series data. In Conference on Causal Learning and Reasoning, pp. 509–525. PMLR, 2022.

[3] Shanyun Gao, Raghavendra Addanki, Tong Yu, Ryan Rossi, and Murat Kocaoglu. Causal discovery in semi-stationary time series. Advances in Neural Information Processing Systems, 36, 2024.

[4] Tijin Yan, Hongwei Zhang, Tong Zhou, Yufeng Zhan, and Yuanqing Xia. Scoregrad: Multivariate probabilistic time series forecasting with continuous energy-based generative models. arXiv preprint arXiv:2106.10121, 2021.

[5] Yan Li, Xinjiang Lu, Yaqing Wang, and Dejing Dou. Generative time series forecasting with diffusion, denoise, and disentanglement. Advances in Neural Information Processing Systems, 35:23009–23022, 2022.

[6] Yusuke Tashiro, Jiaming Song, Yang Song, and Stefano Ermon. Csdi: Conditional score-based diffusion models for probabilistic time series imputation. Advances in Neural Information Processing Systems, 34:24804–24816, 2021.

[7] Paulo HL Rettore, André B Campolina, Artur Souza, Guilherme Maia, Leandro A Villas, and Antonio AF Loureiro. Driver authentication in vanets based on intra-vehicular sensor data. In 2018 IEEE Symposium on Computers and Communications (ISCC), pp. 00078–00083. IEEE, 2018.

---

### Meta-Review · Area_Chair_xXg4 · 2024-12-21

**Metareview:**

This paper introduces a new framework for predicting vehicular CO2 emissions by learning the causal structure of time-series data and performing generative forecasting.
The application is highly compelling and impactful, but there are areas that need improvement, such as the lack of validation for causal discovery on real-world data, better comparisons with other methods, and clearer explanations of the theoretical contributions.

**Additional Comments On Reviewer Discussion:**

The authors sincerely addressed the concerns raised, but their responses were not enough to change the opinions of most reviewers.

---

### Decision · Program_Chairs · 2025-01-22

Reject